# Ocular Inflammation Induced by Immune Checkpoint Inhibitors

**DOI:** 10.3390/jcm11174993

**Published:** 2022-08-25

**Authors:** Florence Chaudot, Pascal Sève, Antoine Rousseau, Alexandre Thibault Jacques Maria, Pierre Fournie, Pierre Lozach, Jeremy Keraen, Marion Servant, Romain Muller, Baptiste Gramont, Sara Touhami, Habeeb Mahmoud, Pierre-Antoine Quintart, Stéphane Dalle, Olivier Lambotte, Laurent Kodjikian, Yvan Jamilloux

**Affiliations:** 1Department of Internal Medicine, Hôpital de la Croix-Rousse, Université Claude Bernard Lyon I, 69004 Lyon, France; 2Research on Healthcare Performance (RESHAPE), INSERM U1290, Université Claude Bernard Lyon I, 69003 Lyon, France; 3Department of Ophthalmology, Bicêtre Hospital, Public Assistance, Hospitals of Paris, Reference Network for Rare Diseases in Ophthalmology (OPHTARA), Paris-Saclay University, 94270 Le Kremlin-Bicêtre, France; 4Department of Internal Medicine & Onco-Immunology (MedI²O), Institute for Regenerative Medicine and Biotherapy (IRMB), Montpellier University Hospital, 34295 Montpellier, France; 5Department of Ophthalmology, Pierre-Paul Riquet Hospital, Toulouse University Hospital, 31300 Toulouse, France; 6Department of Internal Medicine, Le Mans Hospital, 72037 Le Mans, France; 7Department of Internal Medicine, Cornouaille Hospital Center, 29000 Quimper, France; 8Department of Ophthalmology, Nantes University Hospital, 44093 Nantes, France; 9Department of Internal Medicine, Public Assistance, Hospital of Marseille, 13005 Marseille, France; 10Department of Internal Medicine, Saint-Etienne University Hospital, 42055 Saint-Etienne, France; 11Department of Ophthalmology, Pitié-Salpêtrière University Hospital, Sorbonne Université, 75013 Paris, France; 12Department of Pneumology, Eure-Seine Hospital, 27000 Evreux, France; 13Department of Dermatology, Centre Hospitalier Lyon Sud, IMMUCARE, 69495 Pierre-Bénite, France; 14Faculty of Medicine, University Paris Saclay, Public Assistance, Hospital of Paris, 94270 Le Kremlin Bicêtre, France; 15Department of Internal Medicine & Clinical Immunology, Bicêtre Hospital, UMR1184, INSERM, CEA, 94276 Le Kremlin-Bicêtre, France; 16Department of Ophthalmology, Hôpital de la Croix-Rousse, Université Claude Bernard Lyon I, 69004 Lyon, France; 17Lyon Immunology Federation (LIFE), 69000 Lyon, France

**Keywords:** immune checkpoint inhibitors, immunotherapy-related adverse events, uveitis, orbitopathy, eye inflammation, VKH

## Abstract

Ocular immunotherapy-related adverse events (IRAEs), although rare, can be sight-threatening. Our objective was to analyze ocular IRAEs diagnosed in France from the marketing of immune checkpoint inhibitors (ICPIs) until June 2021 and to review the literature. We collected the cases of 28 patients (36 ocular IRAEs), occurring after an average of 17 weeks (±19). Forty-six percent of patients were treated for metastatic melanoma. Anti-PD1 agents were responsible for 57% of the IRAEs. Anterior uveitis was the most common (44%), followed by panuveitis (28%). Of 25 uveitis cases, 80% were bilateral and 60% were granulomatous. We found one case with complete Vogt-Koyanagi–Harada syndrome and one case of birdshot retinochoroidopathy. The other IRAEs were eight ocular surface disorders, one optic neuropathy, and one inflammatory orbitopathy. Seventy percent of the IRAEs were grade 3 according to the common terminology of AEs. ICPIs were discontinued in 60% of patients and 50% received local corticosteroids alone. The literature review included 230 uveitis cases, of which 7% were granulomatous. The distributions of ICPIs, cancer, and type of uveitis were similar to our cohort. Ocular IRAEs appeared to be easily controlled by local or systemic corticosteroids and did not require routine discontinuation of ICPIs. Further work is still warranted to define the optimal management of ocular IRAEs.

## 1. Introduction

There are currently 10 immune checkpoint inhibitors (ICPIs) approved by the U.S. Food and Drug Administration (FDA) and the European Medicines Agency (EMA) [1,2,3]. Each ICPI targets one of four checkpoints (receptors or ligands): Programmed cell Death 1, PD-1 (pembrolizumab, nivolumab, cemiplimab, dostorlimab, and prolgolimab), Programmed cell Death Ligand 1, PD-L1 (atezolizumab, durvalumab, avelumab), cytotoxic T-lymphocyte-associated protein 4, CTLA-4 (ipilimumab), and LAG3 (relatlimab). ICPIs mobilize the immune system to recognize and eliminate tumoral cells [4,5,6]. Since their first use in metastatic melanoma in 2011, they have been evaluated and validated in a growing number of indications, including non-small cell lung cancer, urothelial carcinoma, lymphoma, and clear cell renal cell carcinoma [1].

ICPIs overcome T-cell inhibition to promote tumor cell elimination [4,5]. This results in a loss of immune homeostasis and a facilitation of the inflammatory response. Thus, ICPIs can lead to autoimmune/inflammatory manifestations, remote from the neoplastic site, called immune-related adverse events (IRAEs). IRAEs vary in severity and can involve any organ, but most commonly affect the gastrointestinal tract, followed by the liver, skin, and endocrine system [7]. IRAEs affect 70 to 90% of ICPI-treated patients [8], [9]. Much less frequent, IRAEs affecting the eye have been reported with a frequency ranging from 0.2 to 7.6% [10,11,12,13,14,15,16]. This rarity explains why ocular IRAEs are poorly characterized. Yet, a wide variety of eye disorders have been reported [17,18,19,20] and can be classified into four groups: (i) ocular surface disorders (dry eye disease, blepharitis, conjunctivitis, episcleritis, scleritis, keratitis), (ii) orbital disorders (orbital inflammation, myopathy, and apex syndrome), (iii) uveitis, and (iv) optic neuropathy [21].

While early data came from case reports or small case series [22,23,24,25,26,27], more recent papers are represented by large database analysis [10,14,17,19,20,28,29,30] which include a significant number of cases but do not have a sufficient level of detail to analyze the different eye disorders and their management. The aim of our study was therefore to describe ICPI-induced ocular IRAEs, more specifically the uveitis inflammatory involvement type, detailing the anatomical specificities, potential risk factors, management, and outcome, in relation to the outcome of the underlying tumor. 

## 2. Materials and Methods

### 2.1. Case Identification and Collection

Patients developing ICPI-induced ocular IRAEs were identified through multiple electronic calls for observations to practitioners belonging to (i) the French Society of Ophthalmology (SFO), (ii) the French Society of Internal Medicine, and iii) the French Eye and Internal Medicine Club. Some patients were recruited via the prospective REISAMIC registry (Registre des Effets Indésirables Sévères des Anticorps Monoclonaux Immunomodulateurs en Cancérologie) which was reported previously [16]. REISAMIC is a pharmacovigilance registry which only records grade ≥2 IRAEs and was initiated in June 2014. The study period in REISAMIC was from 1 January 2014 to now. The last case inclusion was in June 2021 and the data export was made in July 2021.

Patients were included if they were aged ≥18 years, and had received an ICPI to treat a solid tumor or hematological malignancy. Patients with concurrent chemotherapy or radiotherapy were not excluded. The diagnosis of ocular involvement had to be confirmed by an ophthalmologist. Patients underwent comprehensive ophthalmologic assessment, as well as ocular and orbital imaging at the discretion of the ophthalmologist. Patients with previous history of ocular disease, with a differential diagnosis (including infections and localized malignancy), or with missing data regarding the type of immunotherapy and ophthalmological description of the IRAE were excluded from the analysis.

For each patient, data were collected from medical records and biological software using an anonymous and standardized electronic Case Report Form (2016 Ennov Clinical, CSOnline v.7.5.720.1, Ennov, Olivier PARIS, Paris, France). Epidemiologic data, medical history, clinical, biological, and imaging data at the time of diagnosis and during the follow-up were collected. Cancer history, neoplastic and ophthalmologic evolution, and treatments received were also recorded. 

### 2.2. Literature Review

The literature review was based on a search for articles that were published before the 31 December 2021. The following PubMed search strategy was performed using the terms (“checkpoint inhibitor” OR “immunotherapy” OR “immune checkpoint inhibitor” OR “ipilimumab” OR “pembrolizumab” OR “nivolumab” OR “atezolizumab” OR “durvalumab” OR “avelumab”) AND (“ocular adverse events” OR “ophthalmologic event” OR “uveitis” OR “VKH (Vogt-Koyanagi-Harada)”). Articles were limited to the English language. Reviews were used to identify potential eligible articles. For uveitis case reports, extracted data included study characteristics (author, publication year), patient demographics (gender, age, cancer type), intervention (ICPI name, potential associated therapies), ophthalmologic outcome (timing of occurrence, evolution), extra-ophthalmologic features (cutaneous, neurological features, neoplastic evolution), and medical care.

### 2.3. Definitions

We used the Standardization of Uveitis Nomenclature (SUN) criteria to classify uveitis [31]. Persistent ocular inflammation corresponded to persistent inflammatory activity at the last ophthalmological visit. Signs of inflammatory activity were the presence of anterior chamber cells (Tyndall), vitreous haze or vitritis, active retinitis on fundus, retinal vasculitis on angiography. The severity of IRAEs was graded according to the common terminology criteria for adverse events [32], version 5.0, November 2017. The diagnostic criteria used were: The Levinson’s criteria [33] or the global diagnostic criteria for birdshot retinochoroidopathy.The international criteria for the diagnosis of sarcoidosis [34]. In the absence of histological proof, we used Abad’s modified criteria [35]. Patients had presumed sarcoid uveitis if they had at least 2 of the following 4 criteria: typical changes on chest X-ray or CT-scan, a predominantly CD4 lymphocytosis on bronchioalveolar fluid analysis, an elevated ACE, or an 18-fluorodeoxyglucose (18-FDG) uptake on scintigraphy. They had indeterminate sarcoid uveitis when only one criteria was met.The revised diagnostic criteria for VKH disease [36]. Complete VKH has to meet the following five diagnostic criteria: (i) absence of history of penetrating ocular trauma, (ii) absence of other ocular disease entities, (iii) bilateral ocular involvement, (iv) neurological/auditory findings, and (v) alopecia, vitiligo or poliosis.

### 2.4. Statistical Analysis

Descriptive analyses are presented as medians (interquartile range, IQR) for non-normally distributed continuous variables and as frequencies and percentages for categorical variables. Patient anonymity was maintained throughout the data collection and statistical analysis phases. All analyses were performed using the statistical software RStudio, v1 3.1093. (R foundation for Statistical Computing, Vienna, Austria).

### 2.5. Ethics

This noninterventional study was conducted in compliance with good clinical practice and the tenets of the Declaration of Helsinki. This study has been authorized by the French National Data Protection Commission (CNIL, Commission Nationale de l’Informatique et des Libertés, Paris, France) and registered under the number 19-157. All patients indicated their non-opposition to the study. Constitution of the REISAMIC registry had been authorized by the French National Data Protection Commission (Commission Nationale de l’Informatique et des Libertés).

## 3. Results

### 3.1. Characteristics of the Study Population

We identified 28 patients who developed ICPI-induced ocular IRAEs. Their age, gender, type of cancer, and ICPI are available in Table 1. The mean age was 59.4 years (standard deviation (SD): 12.2 years) and ranged from 36 to 80. There was no significant age difference by ICPI type. The men-to-women ratio was 1.5.

Seven patients had an ocular medical history: six non-inflammatory ophthalmological histories (two cataracts, one chronic open angle glaucoma, one retinal detachment, one epimacular membrane and one blepharospasm) and one untreated uveitis under dabrafenib, a MAPK inhibitor (patient #11).

Thirteen patients (46.4%) were treated for malignant melanoma, including one for uveal melanoma, six (21.4%) for lung adenocarcinoma, three (10.7%) for pulmonary squamous cell carcinoma, three (10.7%) for a clear cell renal cell carcinoma, one (3.5%) for parotid adenocarcinoma, one (3.5%) for pleural mesothelioma and one (3.5%) for urothelial cancer. All patients had metastatic cancer at the time of ICPI treatment, except for one patient with unresectable squamous cell lung cancer and two patients with unknown metastatic status.

As reported in Figure 1, 16 patients (57%) were treated with an anti-PD-1 monotherapy. Eight (28%) patients were on combination therapy of ipilimumab and nivolumab, one (4%) on atezolizumab, two (7%) on durvalumab and one (4%) was treated with ipilimumab prior to pembrolizumab. One patient treated by nivolumab in January 2018 (i.e., 3 months before she developed uveitis) had been treated with ipilimumab in 2011 and pembrolizumab in 2015 without ocular IRAEs.

### 3.2. ICPI-induced Ocular IRAEs

There were 36 ocular IRAEs among the 28 patients, yielding an average of 1.28 events per patient (Table 2). The mean time to occurrence of all IRAEs combined was 17.6 (±19) weeks. Twenty-one (88%) patients had bilateral involvement at the time of diagnosis. The most common disease was uveitis, with 25 cases present in 22 patients. Eight patients (29%) had ocular surface disorders. Optic neuropathy and orbitopathy were present in one case each (2.8%).

#### 3.2.1. Uveitis

Of the 36 ocular disorders collected, 25 (69.4%) were uveitis, reported in 22 patients (Table 3).

The anatomical distribution of uveitis according to ICPI type is shown in Table 2 and Appendix A in Appendix A. Anterior uveitis was the most common (*n* = 11), followed by panuveitis (*n* = 7), intermediate uveitis (*n* = 4), and posterior uveitis (*n* = 3). Twenty (80%) uveitis were bilateral. Of the 18 uveitis cases involving the anterior segment, 12 (66%) were granulomatous: five had mutton fat keratic precipitates, two had iris nodules, and five had granulomatous features without further details in medical charts. There were six uveitis with synechia, one hypertensive anterior uveitis, one hypertensive posterior uveitis, and one hypertensive panuveitis. All posterior uveitis were chronic and bilateral. One patient had peripheral multifocal choroiditis and papillary edema. One patient had macular edema and one patient had retinal vasculitis.

Patient #9 had bilateral anterior uveitis associated with multiple white spots on the posterior pole. The clinical picture was compatible with a VKH-like uveitis. No long after, she presented with vitiligo, and lymphocytic meningitis, completing the criteria for VKH syndrome.

Three patients had extra-ocular and ocular involvement that suggested a diagnosis of sarcoidosis or birdshot uveitis, for which the diagnostic criteria were not met: Patient #1 presented with bilateral acute granulomatous and synechial anterior uveitis associated with acute intermediate uveitis. She had concomitant hypercalcemia and mediastinal and cervical adenopathy, raising suspicion of sarcoid uveitis. The cervical lymph node biopsy finally revealed metastatic progression of melanoma despite treatment with durvalumab, leading to its discontinuation.Patient #22, who presented with posterior uveitis underwent mediastinal lymphadenopathy biopsy, showing chronic adenitis without evidence of sarcoidosis or tuberculosis.Patient #17, positive for the HLA-A29 antigen, had multifocal choroiditis and optic nerve swelling, without retinal vasculitis, leading to a diagnosis of birdshot-like retinochoroidopathy.

Fourteen (70%) patients had CTCAE grade 3, i.e., anterior uveitis with 3+ or greater cells, intermediate uveitis, posterior uveitis, or pan-uveitis. Three (15%) patients had anterior uveitis with 1+ or 2+ cells corresponding to a grade 2. There was insufficient data regarding the severity of the eye involvement in 3 cases. 

Four patients had other ocular features associated with uveitis: two had scleritis, one did have acute anterior ischemic optic neuropathy and one suffered a dry eye syndrome.

#### 3.2.2. Other Ocular Manifestations

Six (21%) patients had non-uveitic ocular IRAEs, reported in Table 4.

Patient #25 had a perforated corneal ulcer in the context of Sjögren’s syndrome, confirmed by salivary test. The biopsy of minor accessory salivary glands showed borderline histological criteria, without focus; anti-nuclear antibodies were positive (titer, 1/160) without antigen specificity. Despite the discontinuation of nivolumab and local treatment with corticosteroids (eye drops and injections) and oral ciclosporin, he presented repeated rejections of corneal grafts. The histology performed on the enucleation material revealed keratitis with hyperplastic and atrophic areas of the corneal epithelium, intense edema, and neutrophilic infiltrate of Bowman’s membrane stroma. Reactive gliosis was found within the retina and ciliary bodies.

Of the two cases with keratoconjunctivitis, one (the only patient treated with atezolizumab) was complicated by conjunctival fibrosis. 

No ocular surface involvement was found in patients treated by nivolumab + ipilimumab.

Patient #28 had orbital myositis with complete ophthalmoplegia and mydriasis, associated with myocarditis. Myositis-specific antibodies (anti-t-RNA synthetase, anti-MDA-5, anti-TIF1-γ, anti-Mi2, anti-SAE, anti-NXP2, anti-SRP, and anti-HMGCR antibodies), and anti-acetylcholine receptors antibodies were absent. Discontinuation of pembrolizumab and administration of three pulses of i.v. corticosteroids, followed by two courses of i.v. immunoglobulins and methotrexate allowed recovery.

#### 3.2.3. Concomitant Extra-Ocular ICPI-induced IRAEs

Seventeen extra-ocular ICPI-induced IRAEs were reported. The most frequent was vitiligo (*n* = 4), followed by central or peripheral neurological involvement (lymphocytic meningitis, hypertensive meningitis, peripheral neuropathy, cochlear neuritis). The other reported IRAEs were varied and involved one patient each: hepatitis, pancreatitis, colitis, interstitial nephritis, myositis, hypophysitis, hypothyroidism, myocarditis, and lichen planus. One patient (#9) had multiple extra-ocular IRAEs, which were consistent with the diagnosis of a VKH-like syndrome.

### 3.3. Management

The details of ocular management are shown in Table 5. ICPIs were discontinued in 17 (60%) patients, either because of ocular or extra-ocular IRAEs. Three (10%) patients did not receive any specific treatment to treat ocular IRAEs. Nine (32%) patients received only local steroids, while 14 (50%) received systemic steroids; 11 of them (39%) received both. Two patients (#17 and #21) received oral steroids for an extra-ocular indication (pancreatitis and colitis).

#### 3.3.1. Management of ICPI-Induced Uveitis

Thirteen (52%) patients stopped ICPIs after the diagnosis of uveitis (Table 6). In addition to these 13 patients, three (12%) patients under ipilimumab/nivolumab combination therapy discontinued only one of the two ICPIs (ipilimumab). Fifty-four percent of these 16 patients received systemic steroids. Six of the seven patients with panuveitis required systemic steroids. One patient (#15) required several additional immunosuppressive therapies (i.e., methotrexate, abatacept, infliximab, and alpha interferon). Patients #3 and #14 did not receive any specific treatment for ICPI-induced uveitis.

#### 3.3.2. Management of Other Ocular IRAEs

The two patients with keratitis (#25 and #26) and the one with bilateral orbitopathy (#28) required systemic steroids. In addition, patient #28 received methotrexate and i.v. immunoglobulins. Patient #25 was enucleated after seven corneal graft rejections. The three other ocular surface IRAEs were treated with topical steroids.

### 3.4. Outcome

#### 3.4.1. Ophthalmological Outcome

At the end of the follow-up (13.2 months (±11)), 12 (43%) patients had complete ophthalmological remission and 10 (36%) had partial remission. Only one patient (patient #25) had a worsening ocular disorder. Outcome data were not available for three (10%) patients. 

Specifically, the ophthalmological course of uveitis is shown in Figure 2 and detailed for each patient in Table 5. Eighteen (81%) patients had a partial or complete response to treatment. The evolution was unknown for two patients with anterior uveitis. No patient had a worsening of their uveitis at the end of the follow-up. Among the three patients with posterior uveitis, two (66%) patients had partial remission and one (33%) had complete remission. Two (9%) patients had stable uveitis despite treatment: one patient (#6) with subconjunctival steroids, the other (#15) with combinations of intravitreal, topical and i.v. steroids plus other immunosuppressants.

#### 3.4.2. Neoplastic Outcome

Neoplastic evolutions are described in Table 5 and depicted in Appendix A. At the end of the follow-up, nine (32%) patients had stable underlying neoplasia. Ten (36%) patients were in remission, of which seven had a complete remission (five with metastatic melanoma and two with pulmonary adenocarcinoma). ICPIs were maintained in three of these seven patients. Four (14%) patients experienced progression of their neoplastic disease. Among these four patients, ICPIs were discontinued in all four and two received systemic steroids. The neoplastic evolution is unknown for five (18%) patients, but all those with available data, were alive at the last follow-up.

## 4. Literature Review

We identified 84 articles reporting case reports or case series of patients with ICPI-induced ocular IRAEs. Of the 230 reported cases, 143 (62%) had uveitis. Table 7 summarizes the nature of ICPI, the type of uveitis and its management, as well as the outcome of the underlying neoplasm. The mean age at the time of ICPI-induced uveitis was 60.1 (±9.5) years. The men-to-women ratio was 1.9.

The most common uveitis were anterior uveitis (43%) and panuveitis (37%). Of note, half of the patients with panuveitis had features of VKH-like uveitis. Posterior uveitis was the less frequent anatomical type (*n* = 28, 19%).

A total of 96 (67%) patients were treated for metastatic melanoma, followed by non-small-cell lung cancer (13%, 18/143) and renal cell carcinoma (8%, 12/143) (Appendix A). All cancers were metastatic. 

Overall, 76% (109/143) of patients had bilateral uveitis, up to 82% (51/62) in patients with anterior uveitis (Appendix A). Granulomatous involvement was described in only 10 (7%) cases, including five anterior uveitis, one posterior uveitis and four VKH-like uveitis. 22% of uveitis cases were graded 1 or 2 in severity, while 68% were grade 3 or 4. The mean time between ICPI introduction and uveitis diagnosis was 16.8 (±20.3) weeks. Of note, four patients (3%) who had undergone enucleation for the treatment of a uveal melanoma had ICPI-induced uveitis on the remaining eye.

In 88 (61%) uveitis cases, ICPIs were stopped after the onset of uveitis (Appendix A). ICPIs were most frequently discontinued when patients developed posterior uveitis (*n* = 23/28, 82%).

In all, 36 (58%) of the 62 anterior uveitis cases were treated with local steroids only (Appendix A). Among the 26 panuveitis cases, 11 (42%) received only local steroids, and 9 (35%) were treated with a combination of topical and systemic steroids. 

Nivolumab was the most frequent ICPI reported in anterior uveitis (30.6%), VKH-like uveitis (37%), and all types of uveitis (28.6%) (Appendix A). In contrast, pembrolizumab was associated with 32.1% of posterior uveitis and 50% of panuveitis.

Twenty-seven VKH/VKH-like uveitis cases were reported (Table 8) and subdivided into four categories. Melanoma was the most frequently treated cancer (*n* = 19, 70%), followed by non-small cell lung cancer (14%).

Anti-PD1 agents were reported in 77% of patients developing VKH-like uveitis: mostly nivolumab (37%) and pembrolizumab (29%) monotherapies (Appendix A). In 11% of the cases, VKH-like syndrome occurred during ipilimumab/nivolumab combined treatment. 

Overall, 85% of patients received systemic steroids and ICPI were discontinued in 52% of patients (Appendix A).

## 5. Discussion

This study adds 28 new cases of ocular IRAEs induced by ICPIs. Overall, these new cases were similar to those previously reported [14,16,22,25,26,37,41,50]. Indeed, most patients were aged around 60 years, with an overrepresentation of males (ratio = 1.9). Such findings are representative of the epidemiology of cancers treated by ICPIs: melanoma and lung cancers [110,111]. Indeed, almost half of patients had melanoma, which was the first approved indication for ICPIs [13,112], and has already been reported in a previous series [10,11,19,29]. The literature review was also consistent with these findings. Yet, some authors have suggested an alternative hypothesis: melanoma itself would constitute a risk factor for ocular IRAEs, particularly for uveitis [11,26]. This hypothesis is supported by the known relationship between melanoma and ocular disorders or ocular IRAEs [13,14,23,113,114,115,116,117,118,119,120,121,122,123], which seem more frequent in patients treated for melanoma than those treated for other neoplasms. Given these data, one could recommend that symptoms suggestive of ocular-IRAEs should be repeatedly assessed in patients treated with ICPIs for melanoma.

All types of ICPIs were used in our cohort, but none of our patients was treated with ipilimumab as monotherapy. Previous reports have suggested that anti-CTLA-4 could cause more frequent and more serious IRAEs than anti-PD-1 and anti-PD-L1 [124,125,126]. There is no clear explanation for such findings, which remain to be consolidated but previous systematic reviews and meta-analyses have ranked the general safety of ICPIs from low to high as: ipilimumab < pembrolizumab < nivolumab < atezolizumab [127]. In Fang’s study, the anti-CTLA4, ipilimumab had the highest association with uveitis [28] but this finding was conflicting with that of Hou et al., who did not find excess risk for anti-CTLA-4 over anti-PD-1 [128] when looking at all-types ocular IRAEs. As of today, data remain scarce and ICPI prescription is heterogeneous (i.e., anti-PD1 being the most frequently prescribed), precluding definitive conclusions [14,28].

Almost 80% of ocular IRAEs occurred within the first six months after ICPI introduction, which was consistent with the literature [17,27]. Clinicians should thus be aware that ocular IRAEs occur early after ICPI initiation, and alerting symptoms should be explained to patients. As of today, one cannot recommend systematic ophthalmological monitoring since most ocular IRAEs were mild to moderate and treatment-sensitive. However, some late-onset ocular IRAEs have been reported, even after ICPI discontinuation [17]. Information about past ICPI intake should therefore be an integral part of the questioning of ophthalmologists in patients with a history of cancer and unexplained ocular inflammation. 

In our series, almost half the anterior uveitis cases occurred in the setting of nivolumab and or ipilimumab/nivolumab, the combination of which has already been associated with increased rates of anterior uveitis [17]. Bomze et al. have already reported that the addition of an anti-CTLA-4 to an anti-PD1 increases the risk of uveitis, as well as optic nerve disorder and lacrimal disorders [13].

Our study reports a significant number of IRAEs occurring during anti-PD-L1 treatment. This was more rarely reported in the literature, which contains 14 observations of ocular IRAEs under anti-PD-L1, of which seven were uveitis. One explanation is that anti-PD-L1 were the most recently approved and still have limited indications. Interestingly, a significantly higher incidence of posterior uveitis has been described with atezolizumab [17], with some cases of acute macular neuroretinopathy or paracentral acute middle maculopathy with retinal vasculitis [17]. We did not observe such findings in the three patients with posterior uveitis.

Consistent with the literature, uveitis was the most commonly reported IRAE and was bilateral in most cases [17,29,41]. Half of the uveitis were anterior, a frequency that has been reported between 30 and 75% in previous studies [12,13,17,26]. While the majority of our cases of anterior uveitis had granulomatous features, non-granulomatous uveitis predominated in other studies [27,29]. This discrepancy is difficult to analyze but may result from the lack of detailed description of uveitis in case series and case reports.

VKH-like syndromes and VKH-like uveitis seem to have a dedicated place in ocular IRAEs, representing almost half of the panuveitis cases reported in the literature. In their literature review, Dow et al. found that 35% of ICPI-induced panuveitis were VKH-like, suggesting a specific entity [17]. Pathophysiologically, a disruption of the balance between tumor cell killing and immune tolerance to melanocytes has been hypothesized. Similar to our patient, most published cases with ICPI-related VKH-like syndrome had skin involvement 2 to 4 weeks after the first dose of ICPI. This contrasts with typical VKH syndrome in which skin signs occur several weeks or months after the first symptoms (i.e., headaches, uveitis). Most ICPI-related-VKH cases occurred in patients treated for metastatic melanoma, and a few in patients with lung cancer. The relationship between melanoma and VKH-like uveitis is thought to be due to cross reactivity of normal choroidal melanocytes and malignant melanoma cells [129,130], in patients with a possible HLA-related genetic predisposition [131]. Interestingly, the development of VKH has been proposed as a clinical sign suggesting better ICPI efficacy [132]. The complete neoplastic remission of patient #9 is in line with this hypothesis.

Two other phenotypes of ICPI-induced uveitis have been reported: sarcoidosis-like uveitis and birdshot-like chorioretinopathy. Regardless of the ocular involvement, ICPIs may induce other systemic sarcoidosis-like reactions, affecting lymph nodes, lungs or skin [133]. Yet, a history of sarcoidosis does not contraindicate ICPIs [50], and the management is similar to that of sarcoidosis. ICPI discontinuation should only be discussed in case of severe or refractory involvement, and after a careful multidisciplinary evaluation [134]. Neoplastic progression should always be sought as a differential diagnosis (e.g., in case of progressive lymphadenopathy, uveal melanoma).

One patient in our series had birdshot-like chorioretinopathy. This is the second report of such a presentation [84] and both patients received anti-PD-1 in the setting of metastatic melanoma. However, only our patient was positive for the HLA-A29 antigen. It is thus impossible to conclude that HLA-A29 may predispose to ICPI-induced birdshot-like uveitis, nor if the absence of this antigen is sufficient to rule out the diagnosis. Further studies are needed, one axis of which should be the assessment of the predisposing role of the HLA system in ICPI-induced IRAES mimicking HLA-related immune diseases. 

Although they were the second most frequently reported ICPI-induced ocular IRAEs, ocular surface disorders may have suffered reporting bias. Indeed, most of the less invalidating ocular IRAEs (e.g., dry eye, conjunctivitis) may have been underreported, and only severe cases (e.g., keratitis) may have been recorded. In accordance, no isolated dry eye syndrome was recorded, whereas this was the most common ocular side-effect in pivotal RCTs (incidence range, 1.2–24% [8,14,18]. Various ocular surface disorders have been reported, with all ICPIs. Fortunately, most resolved or were controlled with topical treatment (lacrymal substitutes and/or steroid eye drops) [14,23,120,122]. In very rare cases of ulcerative keratitis or corneal perforation, ICPI had to be discontinued [23,122,123,135]. Interestingly, different histological patterns were found in patients with ICPI-induced sicca syndrome compared to patients with Sjögren’s syndrome: increased number of CD3+ T cells, a slight predominance of CD4+ compared with CD8+ T cells, and a paucity of CD20+ B cells in contrast with the immune cell infiltrates of Sjögren’s syndrome [136].

Bilateral papillitis occurred in one patient with aseptic hypertensive meningitis, and was followed by an acute anterior ischemic optic neuropathy. Bilateral non-arteritic ischemic optic neuropathy has been previously reported in only one patient under nivolumab [50] but optic neuritis has been reported in several patients, under various ICPIs (mostly ipilimumab) [137]. Of note, ICPI-associated optic neuritis may present in a more atypical fashion than that associated with multiple sclerosis, with 90% of painless visual loss and/or floaters without vitreous pathology and bilateral optic nerve involvement in 64% [138]. No cases reported the presence of anti-MOG nor anti-aquaporine-4 antibodies.

In our series, as well as in the literature review, most ocular IRAEs were graded CTCAE 3 or 4, whereas in larger studies or RCTs, they were mainly described as mild [9,132]. Here again, there might have been a reporting/publication bias, capturing the most severe cases. Nevertheless, these are the cases more prone to require thorough medical evaluation and systemic treatment.

Most anterior uveitis were treated with topical treatment alone, with a positive outcome [16,19,22,25,26,27,38,41,44,45,48,52,59,63,102]. In the literature, uveitis recurrences have mainly been noted when patients resumed ICPI [17,61] sometimes with a different SUN anatomical classification. In addition, uveitis recurrence was frequently more severe than the first episode. Posterior uveitis were often treated and controlled by steroid intravitreal injections or vitreous implants [22,25,51,64,76,81,103].

ICPIs were discontinued in most patients with ocular IRAEs. The frequency of low grade toxicities, along with the favorable outcome with minimal treatment in most patients, should lead to a reconsideration this attitude, since ICPI discontinuation may not be required in mild to moderate of the cases. The 2019 ASCO guidelines recommended to continue ICPI in cases of grade 1 toxicities, and prescribe artificial tears, as well as referring patients to ophthalmologists within one week [126]. For grade 2 toxicities, they advised to hold ICPI until an urgent ophthalmology referral, and then to administer either topical steroids, cycloplegic agents and/or systemic steroids. Grade 3 and 4 toxicities require definitive ICPI discontinuation, an urgent ophthalmology referral and systemic plus topical steroids; infliximab may be discussed in severe and/or refractory cases [126]. Finally, a multidisciplinary evaluation of the risk/benefit ratio, with experimented oncologists and ophthalmologists, followed by a close monitoring should guide the management of these patients [26].

The limitations of our study include the limited sample size, the retrospective design, and the reporting bias intrinsic to studies based on a call for observations. However, while we have certainly captured the most severe cases, these are the most important ones (i.e., those requiring medical evaluation and management). The design of our study did not allow determining the prevalence of ocular IRAEs. Yet, our study provides detailed case descriptions. Finally, we were not able to collect the doses and administration schemes of ICPIs, precluding the assessment of a dose-dependent effect [28]. Larger cohort studies, such as registries and post-marketing analyses, are needed to better inform clinicians and patients on ICPI-induced ocular IRAEs, mostly regarding their prediction, prevention, and detection, as well as their optimal management.

## 6. Conclusions

Although it captured mostly severe cases, our study is one of the largest reportingl ICPI-induced ocular IRAEs in detail. We note the predominance of anterior uveitis, with an interesting proportion of granulomatous uveitis in our cohort. In addition, uveitis as part of a VKH syndrome is frequently reported with ICPIs and may be an interesting lead for understanding this disease as well as the immunological mechanism of ocular IRAEs. The treatment of ocular IRAEs was mostly simple, with topical treatments, reflecting the high prevalence of anterior segment involvement. Most of the time, ICPIs were discontinued, but this likely reflects a practice that predates the 2019 recommendations and discontinuation of ICPIs is likely no longer warranted. A multidisciplinary approach is required for the management of these patients. Larger studies will soon help us to define the optimal management of these IRAEs.

## Figures and Tables

**Figure 1 jcm-11-04993-f001:**
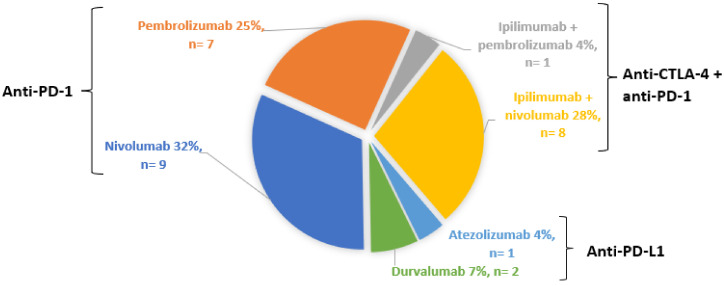
Distribution of ICPIs.

**Figure 2 jcm-11-04993-f002:**
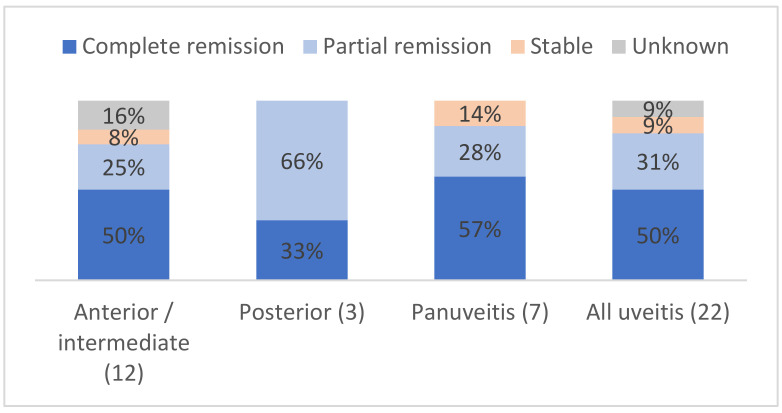
Ocular outcomes depending on the type of uveitis, in %.

**Table 1 jcm-11-04993-t001:** Characteristics of patients with ICPI-induced ocular IRAEs.

	All ICPIs (*n* = 28)	Nivolumab (*n* = 9)	Pembrolizumab (*n* = 7)	Nivolumab + Ipilimumab (*n* = 8)	Nivolumab then Pembrolizumab (*n* = 1)	Durvalumab (*n* = 2)	Atezolizumab (*n* = 1)
Mean age, in years (±SD)	59.4 (±12)	59.5 (±10)	66.3 (±11)	54.4 (±12)	59	59 (±12)	57
Male % (*n*)	60.7 (17)	55.5 (5)	85.7 (6)	50 (4)	100 (1)	50 (1)	100 (1)
Female % (*n*)	39.3 (11)	44.4 (4)	14.3 (1)	50 (4)	0 (0)	50 (1)	-
Indication % (*n*)							
Melanoma	46.4 (13)	44.4 (4)	42.8 (3)	62.5 (5)	100 (1)	-	-
Lung adenocarcinoma	21.4 (6)	22.2 (2)	28.6 (2)	12.5 (1)	-	50 (1)	-
Squamous cell lung cancer	10.7 (3)	22.2 (2)	14.3 (1)	-	-	-	-
Renal cell carcinoma	10.7 (3)	11.1 (1)	14.3 (1)	12.5 (1)	-	-	-
Other	10.7 (3)	-	-	12.5 (1)	-	50 (1)	100 (1)
Mean time to occurrence in weeks (±SD)	17.6 (±19)	25 (±24)	11 (±9)	7 (±6)	65	15 (±1)	34

ICPI = immune checkpoint inhibitor; SD = standard deviation.

**Table 2 jcm-11-04993-t002:** Types of ocular IRAEs according to the ICPI.

	All ICPIs, *n* (%)	Nivolumab, *n*	Pembrolizumab, *n*	Nivolumab + Ipilimumab, *n*	Ipilimumab then Pembrolizumab, *n*	Durvalumab, *n*	Atezolizumab, *n*
All ocular IRAEs	36	13	9	8	2	3	1
Uveitis	25 (69.4%)	9	5	8	1	2	-
- Anterior	11 (30.6%)	3	1	5	1	1	-
- Intermediate	4 (11.1%)	3	-	-	-	1	-
- Posterior	3 (8.3%)	1	1	1	-	-	-
- Panuveitis	7 (19.4%)	2	3	2	-	-	-
- Bilateral	20	7	3	7	1	2	-
Scleritis	2 (5.5%)	-	1	-	1	-	-
Keratitis	2 (5.5%)	2	-	-	-	-	-
Keratoconjunctivitis	2 (5.5%)	-	1	-	-	-	1
Sjögren’s syndrome	3 (8.3%)	1	1	-	-	1	-
Optic neuritis	1 (2.8%)	1	-	-	-	-	-
Orbitopathy	1 (2.8%)	-	1	-	-	-	-

**Table 3 jcm-11-04993-t003:** Detailed ocular findings in patients with ICPI-induced uveitis.

	Gender/Age (Years) Neoplasia	ICPI	Ocular IRAE	B	A	G	S	Initial BCVA	Final BCVA	CTCAE Grade	Onset Time (Weeks)	Other IRAEs
#1	F/47 Lung adenocarcinoma	durvalumab	Anterior and intermediate uveitis with epiretinal membrane	X	X	X	X	OD 1 OS 1	OD 1 OS 1	3	14	/
#2	M/59 Malignant melanoma	ipilimumab; pembrolizumab	Anterior uveitis (Tyndall 2+), scleritis, and bilateral optic nerve swelling	X	X			OD 0.9 OS 0.9	OD 1 OS 1	2	65	Myositis
#3	F/36 Clear cell renal cell carcinoma	ipilimumab + nivolumab	Persistent anterior uveitis	X		X			OD 1 OS 1	1–3	24	Hepatitis
#4	M/77 Pleural mesothelioma	ipilimumab + nivolumab	Anterior uveitis (Tyndall 2+)	X	X			OD 1 OS 1		2	8	Peripheral neuropathy
#5	M/68 Lung adenocarcinoma	ipilimumab + nivolumab	Panuveitis with mutton fat keratic precipitates		X	X			OD 0.8 OS 0.8	3	6	Interstitial nephritis
#6	F/44 Malignant melanoma	ipilimumab + nivolumab	Anterior uveitis (Tyndall 3+)	X				OD 1 OS 1	OD 0.9 OS 0.9	3	3	Hypophysitis
#7	M/52 Malignant melanoma	ipilimumab + nivolumab	Recurrent anterior uveitis with mutton fat keratic precipitate (Tyndall 1+)	X	X	X		OD 1 OS 1	OD 1 OS 1	2	2	Vitiligo
#8	F/52 Malignant melanoma	ipilimumab + nivolumab	Hypertensive posterior uveitis with retinal vasculitis	X				OD 0.7 OS 1	OD 0.8 OS 0.8	3	6	Cochlear neuritis
#9	F/58 Malignant melanoma	ipilimumab + nivolumab	VKH-like syndrome: anterior uveitis with multiple white spots on the posterior pole	X				OD 0.7 OS 0.4	OD 1 OS 1	3	4	Lymphocytic meningitis and vitiligo
#10	M/44 Malignant melanoma	ipilimumab + nivolumab	Anterior uveitis with mutton fat keratic precipitates and anterior Tyndall effect 2+	X	X	X		OD 1 OS 1	OD 1 OS 1	2	3	Vitiligo
#11	M/59 Lung adenocarcinoma	nivolumab	Anterior uveitis	X	X	X	X	OD 1 OS 0.7	OD 1 OS 1	1–3	4	/
#12	M/64 Malignant melanoma	nivolumab	Anterior and intermediate uveitis with iris nodules	X		X	X	OD 1 OS 0.7	OD 1 OS 0.8	3	8	/
#13	F/76 Lung adenocarcinoma	nivolumab	Hypertensive anterior and intermediate uveitis with iris nodules	X	X	X			OD 0.4 OS 0.8	3	45	/
#14	M/64 Malignant melanoma	nivolumab	Unilateral intermediate uveitis with bilateral papillitis and unilateral acute anterior ischemic optic neuropathy		X			OD 1 OS 1	OD 1 OS 0.3	3	8	Hypertensive meningitis
#15	F/46 Malignant melanoma	nivolumab	Acute panuveitis with bilateral choroidal folds and bilateral papillitis	X	X		X	OD 0.3 OS 0.8	OD 0.1 OS 0.1 (cataract)	3	8	/
#16	F/56 Clear cell renal cell carcinoma	nivolumab	Recurrent hypertensive panuveitis with multifocal exudative serous detachment and retinal vasculitis		X	X		OD 1 OS 1	OD 0.7 OS 0.6	3	13	/
#17	F/40 Malignant melanoma	nivolumab	Persistent posterior uveitis with peripheral multifocal choroiditis and bilateral optic nerve edema	X				OD 1 OS 1	OD 1 OS 1	3	13	Pancreatitis
#18	M/45 Clear cell renal cell carcinoma	pembrolizumab	Anterior uveitis and scleritis							1-4	34	Hypothyroidism
#19	M/59 Squamous cell lung cancer	pembrolizumab	Persistent panuveitis with mutton fat keratic, Tyndall 1+, macular and papillary edema	X	X	X	X	OD 1 OS 0.3	OD 0.9 OS 0.8	3	8	/
#20	M/61 Malignant melanoma	pembrolizumab	Panuveitis with mutton fat keratic precipitates, macular edema and dry eye syndrome	X	X	X			OD 0.7 OS 0.4	3	17	/
#21	F/76 Malignant uveal melanoma	pembrolizumab	Panuveitis (vitritis grade 3+)		X		X	OD enucleation OS 1	OD enucleation OS 1	3	5	Colitis
#22	M/73 Malignant melanoma	pembrolizumab	Persistent posterior uveitis with macular edema	X				OD 0.7 OS 0.4	OD 0.6 OS 0.6	3	5	Vitiligo

IRAE = immune-related adverse event; B = bilateral; A = acute; G = granulomatous; S = synechia; BCVA = best-corrected visual acuity (decimal scale); CTCAE = common terminology criteria for adverse events; F = female; M = male; OD = right eye; OS = left eye.

**Table 4 jcm-11-04993-t004:** Detailed description of non-uveitic ocular IRAEs.

	Gender/Age (years) Neoplasia	ICPI	Ocular IRAE	CTCAE Grade	Onset Time (Weeks)	Other IRAEs
#23	F/57 Parotid adenocarcinoma	atezolizumab	Keratoconjunctivitis sicca and conjunctival fibrosis	1–2	34	Lichen planus
#24	M/71 Urothelial carcinoma	durvalumab	Sjögren’s syndrome with bilateral conjunctivitis sicca and unilateral blepharitis; no autoimmunity; normal accessory salivary gland biopsy	1–2	16	/
#25	M/69 Squamous cell lung cancer	nivolumab	Sjögren’s syndrome with unilateral chronic ulcerative keratitis and corneal graft rejection, antinuclear antibodies without specificity	3	78	/
#26	M/62 Squamous cell lung cancer	nivolumab	Keratitis	4	49	/
#27	M/70 Lung adenocarcinoma	pembrolizumab	Keratoconjunctivitis sicca	1	6	/
#28	M/80 Lung adenocarcinoma	pembrolizumab	Bilateral inflammatory orbitopathy with ophthalmoplegia	3–4	6	Myocarditis

**Table 5 jcm-11-04993-t005:** Management and outcomes of ocular IRAES.

Case	Ocular IRAE Type	ICPI Discontinuation	Local Treatment	Systemic Treatment	Ophthalmic Evolution	Neoplastic Evolution
#1	Uveitis	Yes	Steroid eye drops	/	Complete remission	Worsening
#2	Uveitis	Yes	Steroid eye drops	Oral steroids	Complete remission	Stable
#3	Uveitis	Yes	/	/	Unknown	Unknown
#4	Uveitis	Yes †	Steroid eye drops	/	Complete remission	Stable
#5	Uveitis	Yes	Steroid eye drops	Oral steroids 1 mg/kg/d	Complete remission	Stable
#6	Uveitis	Yes †	Subconjunctival steroids	/	Stable	Stable
#7	Uveitis	Yes	Steroid eye drops and lacrimal substitute	/	Complete remission	Partial remission
#8	Uveitis	U	Steroid eye drops	Oral steroids 1 mg/kg/d	Partial remission	Complete remission
#9	Uveitis	Yes	Steroid eye drops	IV and oral steroids	Complete remission	Complete remission
#10	Uveitis	Yes †	Steroid and atropine eye drops	/	Complete remission	Stable
#11	Uveitis	No	Subconjunctival steroids	Oral steroids 0.5 mg/kg/d	Complete remission	Complete remission
#12	Uveitis	No	Steroid eye drops	/	Partial remission	Unknown
#13	Uveitis	U	Unspecified	/	Unknown	Unknown
#14	Uveitis + optic neuropathy	Yes	/	/	(Worsening then) partial remission	Unknown
#15	Uveitis	Yes	Intravitreal steroids and steroid eye drops	IV steroids 500 mg × 3, then oral 1 mg/kg/d methotrexate 0.3 mg/kg/w infliximab 5 mg/kg/m; alpha interferon 180 µg/w abatacept 750 mg/m	Stable	Complete remission
#16	Uveitis	No	Intravitreal steroids, steroid eye drops	Oral steroids 1 mg/kg/d	Partial remission	Stable
#17	Uveitis	Yes	/	Oral steroids 0.5 mg/kg/d (for other IRAEs)	Partial remission	Worsening
#18	Uveitis + ocular surface	Yes	Steroid eye drops	Oral steroids 3 mg/kg/d	Partial remission	Stable
#19	Uveitis	Yes	Subconjunctival steroids, steroid eye drops	Oral steroids 1 mg/kg/d	Partial remission	Stable
#20	Uveitis + ocular surface	No	Steroid eye drops	/	Complete remission	Complete remission
#21	Uveitis	Yes	Subconjunctival steroids, steroid eye drops, atropine eye drops	Oral steroids (for other IRAEs)	Complete remission	Partial remission
#22	Uveitis	No	Steroid eye drops, ketorolac tromethamine	(Topical tacrolimus)	Complete remission	Complete remission
#23	Ocular surface	U	Steroid eye drops and scleral lenses	/	Unknown	Unknown
#24	Ocular surface	Yes	Lacrimal substitute	/	Partial remission	Worsening
#25	Ocular surface	Yes	Subconjunctival steroids, topical ciclosporin 7 corneal grafts, enucleation	Oral steroids 1 mg/kg/d	Worsening	Worsening
#26	Ocular surface	Yes	Ciclosporin eye drops	Oral steroids 1 mg/kg/d	Partial remission	Stable
#27	Ocular surface	Yes	/	/	Partial remission	Complete remission
#28	Orbitopathy	Yes	/	IV steroids 1 mg/kg/d methotrexate 15 mg/w IV immunoglobulins	Complete remission	Partial remission

† nivolumab monotherapy; U = Unknown; d = day; w = week; m = month.

**Table 6 jcm-11-04993-t006:** Management of ICPI-induced uveitis.

	Anterior Uveitis, *n* (%)	Intermediate Uveitis, *n* (%)	Posterior Uveitis, *n* (%)	Panuveitis, *n* (%)	Total of Uveitis, *n* (%)
ICPI management					
- Permanent discontinuation	5 (45)	2 (50)	1 (33)	5 (71)	13 (52)
- Continuation	2 (18)	1 (25)	1 (33)	2 (28)	6 (24)
- Monotherapy ^1^	3 (27)	-	-	-	3 (12)
- Unknown status	1 (9)	1 (25)	1 (33)	-	3 (12)
Topical and systemic steroids	3 (27)	-	1 (33)	6 (86)	10 (40)
Topical steroids only	7 (64)	3 (75)	1 (33)	1 (14)	12 (48)
Systemic steroids only	-	-	1 (33)	-	1 (4)
None	1 (9)	1 (25)	-	-	2 (8)

^1^ Nivolumab monotherapy instead of ipilimumab/nivolumab combination therapy.

**Table 7 jcm-11-04993-t007:** Reported cases of ICPI-induced uveitis.

Type of Uveitis	ICPI	*n*	ICPI Discontinuation	Management (*n* Cases)	Neoplastic Evolution (*n* Cases)	Original Articles
All types	All ICPIs	143				
Anterior	All ICPIs	62				
	atezolizumab	1	N	Local steroids	Progression (1)	[14]
	durvalumab	2	N (2)	Local steroids (2)	Unknown (2)	[27,37]
	ipilimumab	16	N (2) Y (13) Unknown (1)	Local + systemic steroids (6) Systemic steroids (2) Local steroids (8)	Progression (6) Stable (1) Partial response (3) Complete response (1) Unknown (5)	[14,23,38,39,40,41,42,43,44,45,46,47,48,49]
	ipilimumab + nivolumab	15	N (3) Y (9) Unknown (3)	Local + systemic steroids (4) Local steroids (11)	Stable (2) Partial response (4) Unknown (9)	[16,25,26,27,41,50,51,52,53]
	ipilimumab + pembrolizumab	1	M (1)	Local + systemic steroids (1)	Stable (1)	[54]
	nivolumab	19	N (10) Y (7) Unknown (2)	Local + systemic steroids (9) Local steroids (10)	Progression (2) Partial response (8) Unknown (9)	[14,16,25,26,27,37,38,41,55,56,57,58,59,60,61,62]
	pembrolizumab	8	N (2) Y (6)	Local + systemic steroids (3) Local steroids (4) Systemic steroids (1)	Progression (1) Stable (1) Complete response (2) Unknown (4)	[16,22,26,27,63,64,65,66]
Posterior	All ICPIs	28				
	atezolizumab	2	Y (2)	Intravitreal steroids (1) Systemic steroids (1)	Unknown (2)	[22,25]
	cemiplimab	1	Y	Local + systemic steroids	Unknown	[17]
	durvalumab	1	N	Systemic steroids	Unknown	[67]
	ipilimumab	5	Y (4) N (1)	Local + systemic steroids (1) Systemic steroids (3) Local steroids (1)	Progression to death (1) Complete response (1) Unknown (3)	[41,68,69,70,71]
	ipilimumab + nivolumab	3	Y (2) M (1)	Systemic steroids (2) Local steroids (1)	Partial response (1) Unknown (2)	[72,73,74]
	ipilimumab + pembrolizumab	1	Y	Local + systemic steroids	Unknown	[75]
	nivolumab	6	Y (4) N (2)	Intra-ocular steroid implant (1) Local steroids (2) Systemic steroids (1) None (2)	Progression to death (2) Complete response (1) Unknown (3)	[14,76,77,78,79,80]
	pembrolizumab	9	Y (7) N (2)	Local steroids (2) Local + systemic steroids (3) Systemic steroids (1) None (3)	Progression (3) Stable (1) Partial response (2) Unknown (3)	[14,50,78,81,82,83,84,85]
Panuveitis	All ICPIs	26				
	ipilimumab	2	Y (1) Unknown (1)	Local + systemic steroids (1) Local steroids (1)	Unknown (2)	[41]
	ipilimumab + nivolumab	5	Y (2) N (2) M (2)	Systemic steroids (1) Local steroids (4)	Partial response (2) Unknown (3)	[11,22,25,27,86]
	nivolumab	6	Y (4) N (1) Unknown (1)	Local + systemic steroids (2) Topical steroids (2) Systemic steroids (2)	Stable (1) Partial response (1) Unknown (4)	[16,22,41,87]
	Pembrolizumab	13	Y (9) N (2) Unknown (1)	Local + systemic steroids (6) Systemic steroids (2) Local steroids (4) No treatment	Progression (3) Partial response (2) Unknown (8)	[11,14,25,50,66,88,89,90,91]
VKH-like	All ICPIs	27				
	atezolizumab	1	Y	Local + systemic steroids	Unknown	[92]
	ipilimumab	5	Y (3) N (1) Unknown (1)	Local + systemic steroids (2) Systemic steroids (2) No treatment	Stable (1) Partial response (1) Unknown (3)	[24,27,93,94,95]
	ipilimumab + nivolumab	3	Y (2) N (1)	Local and systemic steroids Systemic steroids (2)	Progression (1) Partial response (1) Unknown (1)	[27,96,97]
	nivolumab	10	Y (5) N (1) Unknown (4)	Local + systemic steroids (6) Systemic steroids (2) Local steroids (2)	Progression (1) Partial response (1) Complete response (3) Unknown (5)	[60,98,99,100,101,102,103,104]
	pembrolizumab	8	Y (3) N (2) Unknown (3)	Local + systemic steroids (3) Systemic steroids (4) Local steroids (1)	Partial response (1) Unknown (7)	[22,27,105,106,107,108,109]

M = anti-PD1 monotherapy instead of anti-PD1/anti-CTLA-4 combination; N = no; Y = yes.

**Table 8 jcm-11-04993-t008:** VKH and VKH-like syndrome reported after the use of ICPI.

Type (*n*)	Cancer (*n*)	ICPI (*n*)	Original Articles
Complete VKH syndrome (7)	Melanoma (6) Non-small-cell lung cancer (1)	ipilimumab (3) nivolumab (1) pembrolizumab (3)	[22,93,94,95,98,105,106]
Incomplete VKH syndrome with neurological manifestations (5)	Melanoma (2) Non-small-cell lung cancer (3)	atezolizumab (1) nivolumab (2) pembrolizumab (2)	[92,98,99,107,108]
Incomplete VKH syndrome with cutaneous manifestations (6)	Melanoma (6) whom one choroidal melanoma	ipilimumab + nivolumab (2) nivolumab (3) pembrolizumab (1)	[27,96,100,101,102,109]
VKH-like uveitis (9)	Melanoma (5) Renal cell carcinoma (1) Hypopharyngeal and pharyngeal carcinoma (2) Ovarian cancer (1)	ipilimumab (2) ipilimumab + nivolumab (1) nivolumab (4) pembrolizumab (2)	[22,24,27,60,97,103,104]

## Data Availability

All data relevant to the study are included in the article or uploaded as Appendix A. Additional data may be obtained on reasonable request from the corresponding author.

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
