# Peer review of "Ocular Inflammation Induced by Immune Checkpoint Inhibitors"

_jcm, 2022, doi:10.3390/jcm11174993_

Round 1
Reviewer 1 Report
The manuscript contains a detailed description of the ocular immunotherapy-related adverse events (IRAEs) observed in 28 patients treated with immune checkpoint inhibitors (ICPIs) for metastatic tumors. The major points stemming from such description are the following: a) bilateral uveitis is the most commonly observed IRAE, as previously reported in several case reports, case series and in numerically comparable cohorts (28 patients in the paper by Fortes BH et al [reference 14] and 54 patients in the polycenter study by Thurau S et al [reference 29]); b) male patients are prevalent with a male to female ratio of 1.9/1; c) as expected, malignant melanoma was the tumor most frequently treated with ICPIs given that it is the first approved indication for this class of drugs; d) the anti-CTLA4 ipilimumab seems to be most frequently associated with uveitis, but this finding needs to be confirmed in future studies; e) in the large majority of cases, the IRAEs occur within the first six months from starting the administration of ICPIs; f) the addition of an anti-CTLA-4 to an anti-programmed cell death protein-1 [PD1] augments the risk of uveitis; g) IRAEs can also occur during the administration of anti-PD-L1 agents.
The topic is of interest in consideration of the increasing number of neoplastic patients who are treated with ICPIs and is therefore worth of publication. However, as many as 17 French authors (an objectively high number partially justified by the polycenter nature of the study), 16 pages of text and 138 references have been employed to convey these messages. The present reviewer feels that the manuscript would greatly improve by the avoidance of redundancy and obviousness, as well as of repetitions in the text of points already mentioned in the tables. Here are a few examples:
- Page 3, lines 100-101: “Ocular IRAEs were only considered if they did not precede exposure to ICPI”. Pleonastic statement.
- Page 4, lines 183-185: “No patient was treated with cemiplimab,….”. These ICPIs are in fact not mentioned in Table1. Again, pleonastic statement.
- Page 5, lines 188-195: this information is all reported in Table 2.
- Page 10, line 297 and line 303 report the same information concerning enucleation.
- Table 5 and Figure 3 substantially summarize the same points related to the neoplastic outcome.
- Page 14, lines 360-363: the four categories of VKH are already reported in Table 8.
The following inconsistencies should be corrected:
- Page 4, paragraph 170-176 compared with Table 1: the decimals are present in the Table and omitted in the text or viceversa.
- The acronym ICPI should be explained in extenso as footnote to Table 1 (where it is absent) and not repeated in the other Tables.
- In the title of Table 1, IAREs should read IRAEs.
- Orbitopathy is reported in 2.8% in the text (page 5, line 195) and in 2.7% in Table 2.
- Page 7, lines 211-212: ”There were six uveitis with synechia”; five patients with synechiae can instead be counted.
- Page 7, lines 222-226: the characteristics of patient #2 do not correspond to those reported in Table 3.
Minor points:
- The acronyms PD1, VKH, MAPK should be given in extenso when written the first time.
- “ophthalmic” and “systematic review” are too generic terms to be considered keywords for the specific topic of this manuscript.
- Page 8, Table 4 and line 268: lichen planus; oral? Genital? Other?
- Page 8, line 259: “Myositis-specific antibodies”: please, specify their corresponding antigen, if tested.
- Page 9, lines 277-278: should pancreatitis be treated with oral steroids?
- A few typos need to be corrected:
a) Title of Table 3: “ccular” should read “ocular”;
b) Page 10, line 305: “detailled” should read “detailed”;
c) Page 13, line 340: “cancer” should read “cancers”;
d) Page 17, line 507: “these are the certainly…” should read “these are certainly…”.
Author Response
Dr Yvan JAMILLOUX
Internal Medicine Dpt
Hôpital de la Croix-Rousse
103, grande rue de la Croix-Rousse
69317 Lyon CEDX 04
France
(+33) -4.26.73.26.31
yvan.jamilloux@chu-lyon.fr
Dear editor and dear Jiaying Chen,
Please find enclosed the revised manuscript entitled “Ocular inflammation induced by immune checkpoint inhibitors”.
To avoid the redundancy that you highlighted, we deleted these repetitions in the text:
- those suggested (page 3, lines 100-101, page 4, lines 183-185; page 5, lines 188-195; page 10, line 303; page 14, lines 360-363)
- names of ICPI in page 2, line 96;
- details page 4, line 180, already reported in figure 1;
- Figure 3 belongs now to supplementary data (Figure S2).
Supplementary information asked, such as the localisation of lichen planus (page 8, line 270) remains unfortunately unknown. Myositis-specific antibodies were detailed (page 8, line 261).
Furthermore, although the contribution of corticosteroids to the treatment of autoimmune pancreatitis remains uncertain1, the recommendations in 2020 called for corticosteroids and the suspension of ICPI2. Patient #17 therefore received corticosteroids for this indication (page 9, line 280), which may now be debated.
We corrected the typographical errors and inconsistencies reported as well as the number and names of approved ICPI (page 2, line 54 to 60).
Information about patient #19 were corrected: this case was the one missing among the six synechial uveitis reported.
We hope the revisions made to this manuscript will be sufficient. We look forward to hearing from you soon.
Kinds regards,
On behalf of the co-authors,
Dr Florence CHAUDOT and Dr Yvan Jamilloux
1Sayed Ahmed, Ahmed, Michael Abreo, Anusha Thomas, et Suresh T. Chari. « Type 3 Autoimmune Pancreatitis (Immune Checkpoint Inhibitor-Induced Pancreatitis) ». Current Opinion in Gastroenterology 38, no 5 (septembre 2022): 516‑20. https://doi.org/10.1097/MOG.0000000000000873.
2Thompson, John A., Bryan J. Schneider, Julie Brahmer, Stephanie Andrews, Philippe Armand, Shailender Bhatia, Lihua E. Budde, et al. « NCCN Guidelines Insights: Management of Immunotherapy-Related Toxicities, Version 1.2020 ». Journal of the National Comprehensive Cancer Network 18, no 3 (mars 2020): 230‑41. https://doi.org/10.6004/jnccn.2020.0012.

Reviewer 2 Report
There are minor formatting/spacing errors that should be corrected. Science, organization, and construction of the review are good.
Author Response
Dr Yvan JAMILLOUX
Internal Medicine Dpt
Hôpital de la Croix-Rousse
103, grande rue de la Croix-Rousse
69317 Lyon CEDX 04
France
(+33) -4.26.73.26.31
yvan.jamilloux@chu-lyon.fr
Dear editor,
Please find enclosed the revised manuscript entitled “Ocular inflammation induced by immune checkpoint inhibitors”.
We corrected the typographical errors (title of Table 1, “IRAEs”; title of Table 3, “Ocular”; page 10 line 305 “detailed”; page 13 line 340 “cancers”; page 17 line 507 “these are certainly”) and some inconsistencies (table 2, “orbitopathy and optic neuropathy: 2.8%”; the number and names of approved ICPI (page 2, line 54 to 60)).
To avoid the redundancy highlighted by you colleague, we deleted these repetitions in the text:
- those suggested by your colleague (page 3, lines 100-101; page 4, lines 183-185; page 5, lines 188-195; page 10, line 303; page 14, lines 360-363)
- names of ICPI in page 2, line 96;
- details page 4, line 180, already reported in figure 1;
- Figure 3 belongs now to supplementary data (Figure S2);
- We explained ICPI, IRAE as footnote to Table 1 and did not repeated in other Tables.
We added some information :
- Myositis-specific antibodies were detailed (page 8, line 261).
- Information about patient #19 were corrected: this case was the one missing among the six synechial uveitis reported.
We hope the revisions made to this manuscript will be sufficient. We look forward to hearing from you soon.
Kinds regards,
On behalf of the co-authors,
Dr Florence CHAUDOT and Dr Yvan Jamilloux
Reviewer 3 Report
Dear Authors,
Thank you for this valuable, original and well-written manuscript.
ICPI IRAEs are well known side effects in general; however, not so much is known about ICPI ocular IRAEs (when compared to others).
On top of that, practical recommendations and overview regarding ICPI ocular IRAEs are still under-represented in ESMO (European Society of Medical Oncology) and NCCN (National Cancer Comprehensive Network) guidelines, mostly due to lack of scientific data (predominantly based on individual case reports/series) and their relatively low occurrence.
To the best of my knowledge, this study is one of the largest reporting in detail ICPI-induced ocular IRAEs. Thus, it is of high importance since it provides really valuable insights into the latter field.
Methodology is adequate, findings are clearly and comprehensively presented, review of literature is provided, all relevant (methodological) limitations are highlighted, and conclusions are supported by the results.
Best regards, Reviewer
Author Response

(The authors gave the same response as above.)
